

# Revisiting ice sheet mass balance: insights into changing dynamics in Greenland and Antarctica from ICESat-2

Nicolaj Hansen[1,2], Louise Sandberg Sørensen[1], Giorgio Spada[3], Daniele Melini[4], Rene Forsberg[1], Ruth Mottram[2], and Sebastian B. Simonsen[1]

[1]Geodesy and Earth Observation, DTU-Space, Technical University of Denmark, Lyngby, Denmark
[2]Danish Meteorological Institute, Copenhagen, Denmark
[3]Dipartimento di Fisica e Astronomia (DIFA) "Augusto Righi", Alma Mater Studiorum Università di Bologna, Bologna, Italy
[4]Istituto Nazionale di Geofisica e Vulcanologia, Roma, Italy

**Correspondence:** Nicolaj Hansen (nichsen@space.dtu.dk)

**Abstract.** The time series of observations from NASA's latest satellite laser altimetry, the Ice, Cloud, and Land Elevation Satellite-2 (ICESat-2) are now mature to revisit the methodology for estimating surface elevation change and mass balance of ice sheets as proposed by Sørensen et al. (2011). Following the original ICESat study, we combine the derived ICESat-2 surface elevation change estimates with modelled changes of both the firn and the vertical bedrock to derive the total mass balance of the ice sheets, during the northern hemisphere mass balance years of October 2018 to September 2021. The method of converting the surface elevation change to mass balance change has been refined to obtain more reliable mass balance results for both ice sheets. From 2018 to 2021, we find that the grounded ice sheet in Antarctica has lost 135.7±27.3 Gt year[-1], and the Greenland ice sheet 237.5±14.0 Gt year[-1]. Compared to 2003-2008, the ICESat-2 derived mass change of the Greenland ice sheet has a similar magnitude; however, the spatial pattern is changed and we observe reduced ice loss around Jakobshavn Isbræ and in the southeast accompanied by increased loss almost everywhere else and especially in the northern sector of the ice sheet. Our results show pervasive ice sheet loss across much of Greenland in recent years and an increase in loss from Antarctica compared to earlier studies. Parallels between the two ice sheets revealed by ICESat-2 data reflect atmospheric and oceanic drivers and show the importance of understanding ice sheets as components within the Earth system.

## 1 Introduction

The Greenland ice sheet (GrIS) and the Antarctic ice sheet (AIS) are the two largest ice bodies on Earth. In spite of their contrasting geographical contexts, both have experienced an acceleration in ice loss (negative mass balance) in recent decades (Shepherd et al., 2018, 2020; Otosaka et al., 2023), and have become the largest mass-component contributor to global sea-level rise (Horwath et al., 2022). Changes in the mass balance of both ice sheets reflect atmospheric and oceanic drivers. The first accelerations of Antarctic ice loss were seen in the West AIS (WAIS) in the 1980s (Forsberg et al., 2017; Rignot et al., 2019),



and attributed to warm circumpolar deep water melting the floating glaciers from below, thus resulting in an increase in ice discharge (Velicogna and Wahr, 2006a; Pritchard et al., 2012; Paolo et al., 2015; Wåhlin et al., 2021). The WAIS is especially vulnerable to rapid retreat triggered by increasing ocean temperatures due to its retrograde bedrock at the grounding line that could lead to marine ice sheet instability (Mercer, 1978; Morlighem et al., 2020). Consequently, much research has focused on

understanding mass changes in this region given the possibility of rapid sea level rise. Conversely, for most of the observational period, the East AIS (EAIS) was considered a region of ice gain (positive mass balance) (Smith et al., 2020; Wang et al., 2021) or else close to equilibrium (Davis et al., 2005; Shepherd et al., 2018; Bamber et al., 2018; Rignot et al., 2019), due to enhanced accumulation. In-situ observations, as well as climate reanalysis, show that parts of the mass changes in coastal East Antarctica and the Antarctic Peninsula have been attributed to multidecadal patterns of variability, including a long-term positive trend in

the Southern Annular Mode (SAM) (Marshall et al., 2017; Fogt and Marshall, 2020). Extreme snowfall events for example in Dronning Maud Land in 2013, attributed to the landfall of an atmospheric river (Wille et al., 2021) can also affect mass balance on annual to multi-annual timescales. These features can also be assessed via trends in surface altimetry. Recent studies show however that parts of the EAIS, such as Wilkes Land and Oates Land are also experiencing a net loss of ice (Gao et al., 2019; Velicogna et al., 2020; Stokes et al., 2022) which may be related to oceanic drivers and the ice dynamical response. This can

be assessed by combining altimetry and firn modelling (Pelle et al., 2020; McCormack et al., 2021; Verjans et al., 2021).

In the northern hemisphere, GrIS was observed to be close to equilibrium in the 1990s transitioning to net mass loss in the early 2000s which accelerated up through the 2000s (Velicogna and Wahr, 2006b; Velicogna, 2009; Sørensen et al., 2015; Simonsen et al., 2021). On a regional scale, mass loss rates show high interannual and decadal variability, but the GrIS has experienced mass loss along most of its western and southeastern margin (Mottram et al., 2019; Shepherd et al., 2020) with

Jakobshavn Isbræ as the single largest outlet for mass loss (Joughin et al., 2008; Enderlin et al., 2014). Recent studies have detected a slow-down of the Jakobshavn Isbræ (Khazendar et al., 2019; Joughin et al., 2020), which according to Khazendar et al. (2019) is likely due to a small cooling of the ocean water in Disco Bay, though other outlet glaciers, particularly in southeast Greenland (Helheim Glacier) increased their discharge during the same period (Mankoff et al., 2020). An increase in mass loss in the northern part of the GrIS in the late 2010s emerges from both increases in runoff and discharge (Mouginot et al.,

2019; Black and Joughin, 2022), partly related to changes in atmospheric circulation patterns (Noël et al., 2019), the Greenland blocking index (Hanna et al., 2016) as well as extreme events like atmospheric rivers in northwest Greenland which lead to strong foehn winds in north-eastern Greenland, driving intense melt (Mattingly et al., 2018, 2023). Atmospheric circulation changes partly modulate precipitation but are also associated with changes in clouds that affect the surface energy and mass balance. Noël et al. (2019) found that the northernmost part of the GrIS has experienced a large and rapid expansion of the

ablation zone, due to changes in the atmospheric circulation, which increased cloud cover in the early summer and enhanced atmospheric warming through decreased longwave heat loss. A similar process may be behind the observed melt events of the Marie Byrd Land (Scott et al., 2019; Zou et al., 2021) and over the Antarctic Peninsula (Datta et al., 2019).

The continuing net negative mass balance of both ice sheets, and the observed changes on a regional scale, have been closely linked to anthropogenic climate change accelerating existing weather patterns (Masson-Delmotte et al., 2021). It is societally

extremely important to understand these processes and their likely future rates, particularly in terms of planning for sea level



rise adaptation in low-lying regions (Jevrejeva et al., 2018; Hauer et al., 2020). Given that both ice sheets are affected by the same driving processes in the atmosphere and the ocean, the more recent and rapid changes in Greenland likely point to the near future evolution of Antarctica as it transitions to a warmer climate. In this study, we therefore apply satellite altimetry to assess recent changes in ice sheet geometry and use insights from the application of numerical models to interpret these changes in the light of observed atmospheric, ocean, and ice dynamical processes.

Satellite altimetry is used to monitor the surface elevation change (SEC) of the ice sheets to derive mass balance on the basis of glaciological assumptions. This geodetic method has been used in previous studies for both ice sheets (Sørensen et al., 2011; Zwally et al., 2011, 2015; Shepherd et al., 2018, 2020; Simonsen et al., 2021; Khan et al., 2022). These studies show that the integrated mass loss from GrIS contributed 10.8±0.9 mm of global mean sea level rise, in the period from 1992 to 2018 (with more than 80% of it after 2003) (Shepherd et al., 2020), while AIS contributed 7.6±3.9 mm of global mean sea level rise from 1992 to 2017 (Shepherd et al., 2018). Here, we aim to further extend the long time series using data from the Ice, Cloud, and Land Elevation Satellite-2 (ICESat-2) to estimate SEC for both AIS and GrIS. We then use numerical models to calculate the non-ice mass-related signals from changes in the near-surface firn, as well as the vertical movement of the bedrock to update the density parameterization needed for converting the corrected volume change into mass change over both ice sheets. The new 2018-2021 GrIS mass balance can be compared to mass change estimates derived by ICESat during the period 2003-2008 (Sørensen et al., 2011) to get insights into the spatial changes of mass loss in Greenland over time and thus get a better understanding of the GrIS response to continued global warming.

## 2 Data and Methods

Our methodology closely follows the repeat track method approach used by Sørensen et al. (2011, (Method 3)) for ICESat data and we therefore refer to this paper for methodological details of the analysis. However, we highlight improvements introduced in the ICESat-2 processing chain and updates to the model assumptions made for consistently deriving mass balance for both ice sheets.

### 2.1 Surface Elevation Change and ICESat-2

ICESat-2, the follow-on mission to ICESat (Smith et al., 2019a) was launched in September 2018, into a near pole orbit (92°inclination) and an orbital path with a repeat period of 91 days. ICESat-2 is carrying the ATLAS (Advanced Topographic Laser Altimeter System) instrument. ATLAS is based on a green laser (532 nm, in contrast to the red laser of the original ICESat), which is split into three beam pairs separated by approximately 3.3 km on the ground, with each pair separated by about 90 m. This six-beam array makes it possible to measure small-scale across-track surface slopes, further helped by the receiver being a photon-counting instrument, which times the return of the reflected laser beam at the photon level, resulting in an integrated footprint of about 17 m on the ground (Neumann et al., 2019).

Here, we use the ATL06 release 005 data product, which provides the averaged geolocated photon heights at a 40 m posting (Smith et al., 2019a, b). The SEC from ICESat-2 is derived using the repeat track method, which performs a least squares





regression on all data in 5000 m along-track segments of the satellite track. The function used in the regression is given by:

$$H(x,y,t) = H_0(\bar{x},\bar{y}) + \frac{dH}{dt}(t-\bar{t}) + sx(x-\bar{x}) + sy(y-\bar{y}) + \alpha\cos(\omega t) + \beta\sin(\omega t) + \epsilon(x,y,t) \tag{1}$$

where the parameters to be estimated in the regression $H(x,y,t)$ is the surface elevation at time $t$ and position $x$ and $y$, $H_0$ is
the mean elevation, and $sx$ and $sy$ describe the surface topography by its slope. $\alpha\cos(\omega t) + \beta\sin(\omega t)$ is the seasonal signal,
and $\epsilon$ is the residual between the model and the data, the overbars indicate the mean of the measurements in a segment. For
further details we refer to (Sørensen et al., 2015, 2018). The SEC from ICESat-2 is shown in Fig. 1 for both GrIS and AIS. The
ICESat-2 data are then interpolated geostatistically using ordinary kriging. In this process, an exponential variogram model
with a range of 50 km is employed, derived from all available data. The choice of range and model is determined from the
experimental variogram, the full method can be found in Sørensen et al. (2011) and we use the implementation in the R package
gstat (Pebesma, 2004). The results after ordinary kriging has been applied are shown with the associated errors in Fig. 2. The
observed SEC signal can be related to the glaciological components of an ice sheet and can be written as follows:

$$\frac{dH}{dt} = \frac{\dot{b}}{\rho} + w_c + w_{ice} + \frac{\dot{b}_m}{\rho} + w_{br} - u_s\frac{dS}{dx} - u_b\frac{dB}{dx}, \tag{2}$$

In Eq. 2 $H$ is the surface elevation, $\dot{b}$ is the surface mass balance, $\rho$ is the density of ice or snow, $w_c$ is the firn compaction
velocity, $w_{ice}$ is the vertical ice velocity, $\dot{b}_m$ is basal mass balance, $w_{br}$ is the vertical bedrock velocity, $u_s$ is the horizontal
velocity of ice at the surface $S$ and $u_b$ is the horizontal velocity of ice at the bed $B$ (Sørensen et al., 2011). The SEC grids are
also corrected for non-ice mass-related elevation change, i.e. firn compaction and bedrock movement as described in detail in
the following sections.


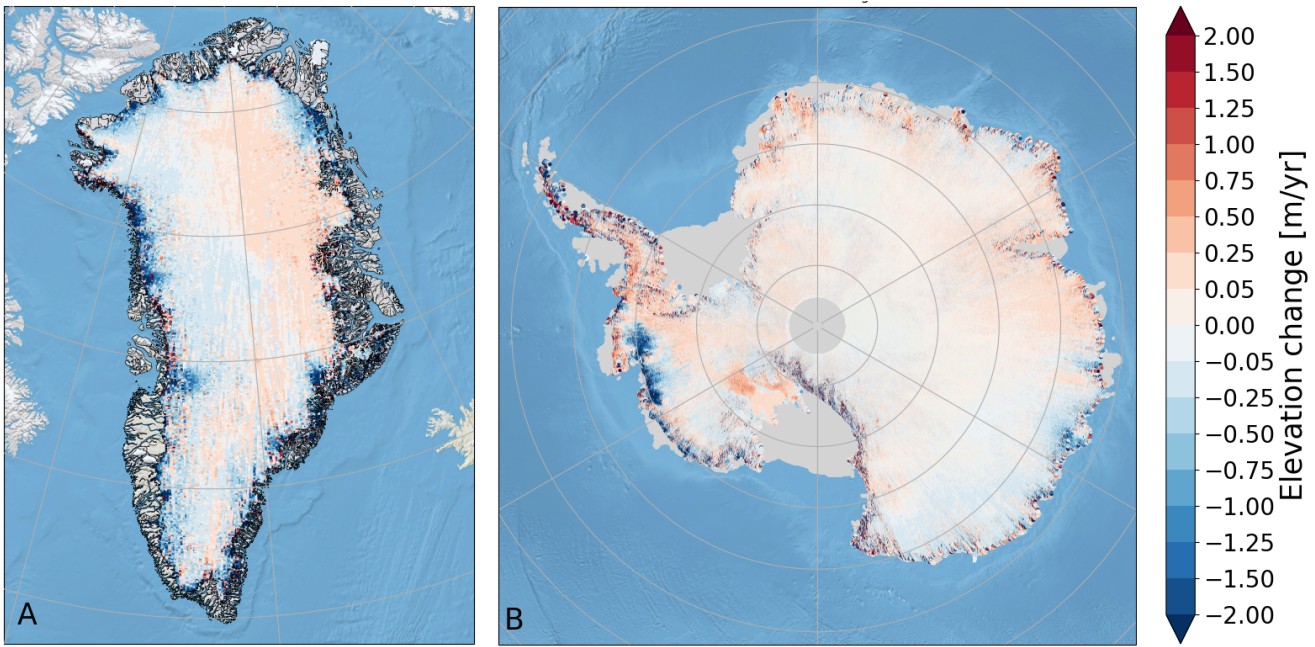

**Figure 1.** Ice sheet elevation change from October 2018 to September 2021 derived from ICESat-2 observations for Greenland (A) and Antarctica (B), before applying the kriging algorithm. The gray area around Antarctica represents the ice shelves (Gerrish et al., 2021). The polar gap can be seen at the centre of the AIS. Units are in metres of ice equivalent per year.



**Figure 2.** Ice sheet elevation change (A, B) from October 2018 to September 2021 derived from ICESat-2 observations and associated uncertainties (C, D), after applying the kriging algorithm. Units are in metres of ice equivalent per year.



## 2.2 Firn correction

To correct for changes in surface elevation due to firn compaction, we use an offline surface energy and firn model. The model is forced by the atmospheric regional climate model (RCM) HIRHAM5 and simulates the physics within the firn. The firn model is described in Langen et al. (2015, 2017) and Hansen et al. (2021), and includes a sophisticated snow and ice scheme that handles wet and solid accumulation, melt, retention, and refreezing of liquid water, runoff, and compaction. The six hourly forcing from the RCM HIRHAM5 includes snowfall, rainfall, evaporation, sublimation, net sensible and latent heat fluxes, and downwelling shortwave and longwave radiative fluxes. In this study, HIRHAM5 is forced on the lateral boundaries with the ERA-5 reanalysis dataset (Hersbach et al., 2020) and dynamically downscaled to $0.05°(\approx 5$ km) resolution for Greenland and to $0.11°(\approx 12.5$ km) in Antarctica. HIRHAM5 was first documented in Christensen et al. (2007) and evaluated over Greenland (Lucas-Picher et al., 2012; Langen et al., 2017) and over Antarctica (Hansen et al., 2021; Mottram et al., 2021; Orr et al., 2023). The HIRHAM5 forced surface mass balance is comparable with other RCMs over both ice sheets (Fettweis et al., 2020; Mottram et al., 2021).

The compaction of firn changes the ice sheet elevation of the column without changing its mass. To correct for this effect we use the outputs from the firn model to compute the firn air content $(dh_{air})$ for each month, as we seek non-mass related changes:

$$dh_{air} = \sum_{i=1}^{N_{30}} \left(1 - \frac{\rho_{bulk_i}}{\rho_{ice}}\right) H_i, \tag{3}$$

In Eq 3, the index $i$ runs over the layers, $N_{30}$ is the number of firn layers within a column that is necessary to reach a depth corresponding to 30 years of precipitation. $\rho_{bulk_i}$ is the density of the $i$-th layer of the firn model including ice lenses, $\rho_{ice} = 917$ kg m$^{-3}$ is the ice density and $H_i$ is the depth of the $i$-th layer. We then calculate the trend in the monthly $dh_{air}$ by applying a linear fit to Eq. 3 that accounts for seasonal changes and thus derive the trend in compaction $\left(\frac{dh_{air}}{dt}\right)$. Over both ice sheets, HIRHAM5 has been run from 1980 to 2021, so by setting the depth to 30 years of model precipitation, instead of the full depth of the firn pack we can avoid biases introduced from the spin-up period. However, in eastern Antarctica, due to the low precipitation rates, a longer spin-up was necessary, and there a minimum of ten layers has been used where the 30 years of precipitation threshold was inadequate. Figure 3 (A, B) shows the resulting firn compaction rates for GrIS and AIS respectively.

## 2.3 Bedrock uplift correction

We estimate the rate of vertical bedrock uplift by modelling the contributions due to *i)* the still ongoing glacial isostatic adjustment (GIA) in response to the melting of the late-Pleistocene ice sheets and *ii)* the elastic rebound (ER) processes due to present-day ice mass change.

The GIA contribution to the present-day vertical velocity is evaluated by computing a numerical solution of the gravitationally and topographically self-consistent Sea Level Equation (SLE), which accounts for deformational, gravitational, and rotational effects induced by the spatiotemporal evolution of ice and meltwater loads. The numerical solution has been obtained



by means of the open-source SELEN[4] solver (Spada and Melini, 2019), in which the ICE-7G(VM7) ice sheet chronology (Roy and Peltier, 2015, 2017) has been implemented. The SLE has been solved on a global icosahedral grid with a spatial resolution of $\sim 40$ km and including harmonic terms up to $L_{max} = 512$, which by Jeans' rule corresponds to a minimum wavelength of $\sim 80$ km on the Earth's surface. The rotational feedback has been accounted for following the revised theory of Mitrovica et al. (2005) and Mitrovica and Wahr (2011). The numerical solution scheme for the SLE involves internal iterations, in which the solution is progressively refined for a given paleo-topography, and external iterations in which the topographic evolution is updated according to the estimates of relative sea-level change. To ensure convergence, we performed five internal iterations and five external iterations. To prescribe the present-day global topography, we employed the bedrock version of the ETOPO1 global topographic model (Amante and Eakins, 2009), integrated with the Bedmap2 relief (Fretwell et al., 2013) below $60°$S latitude.

To model the contribution to the vertical velocities stemming from the ER in response to present-day ice melt, we employed a suitably adapted version of the REAR code (Melini et al., 2015). The mass balance has been discretized into disc-shaped ice elements having a half-amplitude of $\sim 2.8$ km. The elastic response to a unitary mass change acting on a single disc element was computed analytically according to the solution described by Bevis et al. (2016), employing a set of elastic loading Love numbers associated with the seismological REF Earth model and including harmonic terms up to degree $L_{max} = 40,000$, ensuring an accurate spectral representation of spatial scales down to $\sim 1$ km (Kustowski et al., 2007). The contributions of all the disc elements describing the mass balance model have been combined numerically taking the uncertainties associated with the input mass balance into account. The vertical bedrock movement, which is the sum of GIA and ER, over the two ice sheets is displayed in Fig. 3E and F.

To evaluate the ER contribution to the vertical velocities, we use a first estimate of the mass balance in which the correction for bedrock uplift has not been taken into account. Then, the modelled elastic rates are used to update the mass balance by applying the bedrock uplift correction. One iteration of this process were shown sufficient to ensure convergence of the mass balance (Sørensen et al., 2011).

## 2.4 Volume to mass conversion

We correct the observed elevation changes for signals that are not associated with a mass change, e.g. firn compaction rate and the vertical bedrock movement. We convert the corrected SEC, $\left(\frac{d\tilde{H}}{dt}\right)$, to mass change by the appropriate density, $\tilde{\rho}$:

$$\frac{dM}{dt} = \frac{d\tilde{H}}{dt}\tilde{\rho}, \tag{4}$$

where $\frac{d\tilde{H}}{dt}$ results from a change in either melt, snow accumulation, or dynamical behavior, meaning that $\tilde{\rho}$ is dependent on which physical processes drive the observed SEC. The choice of appropriate densities is associated with great uncertainties and impacts the resulting mass balance estimates (Shepherd et al., 2012). A positive SEC, here defined as the surface moving upwards $\left(\frac{d\tilde{H}}{dt} > 0\right)$, will be due to snow accumulation in areas with negligible changes in ice dynamics, making $\tilde{\rho} = 250\text{-}400$ kg m$^{-3}$. Whereas a negative SEC, here defined as the surface moving downwards $\left(\frac{d\tilde{H}}{dt} < 0\right)$, can be assumed to be caused by





surface melt or ice dynamics; in which case $\tilde{\rho}$ will be 917 kg m$^{-3}$. Several parameterizations for $\tilde{\rho}$ have been published in earlier
studies (Thomas et al., 2006; Sørensen et al., 2011; Zwally et al., 2011, 2015, 2021). Based on ICESat observations, Sørensen et al. (2011) observed a predominantly positive SEC signal for the GrIS above the equilibrium line altitude (ELA). They assumed all positive changes in the accumulation zone were due to snow accumulation. Conversely, negative changes, mostly observed in the ablation zone (below the ELA), hence assumed to be due to either melt or ice dynamics. This assumption was robust in most cases, as they had taken the signals from the firn compaction and the vertical bedrock movement into account. Other studies showed that the interior of GrIS experienced a thickening from the early 1990s through the 2000s due to increasing snowfall (Thomas et al., 2006; Broeke et al., 2009). Furthermore, dynamical build-up of ice is scarce in Greenland and generally happens below the ELA (Pritchard et al., 2009). However, our results show small negative SEC signals in our in $\frac{d\tilde{H}}{dt}$, both in the interior of the GrIS and close to the South Pole in Antarctica, where the horizontal ice velocity is very low (< 4 metre per year), hence we do not believe that it is physically plausible that this widespread negative SEC is due to ice dynamics. Moreover, around the Siple Coast (WAIS), the Kamb Ice Stream has stagnated and the Mercer and Whillans Ice Streams have been decelerating, leading to a positive mass balance in this region due to a dynamical build-up of ice (Shepherd et al., 2019). This build-up of ice is due to spatial variability in the ice flow velocities, as the bedrock near the Siple Coast has higher friction than the bedrock inland, thus slowing down the ice flow (Anandakrishnan and Alley, 1997; Scheuchl et al., 2012). Furthermore, over the grounded AIS large parts of the interior show very small magnitudes in the SEC signals that are functionally 0 when the uncertainty range is taken into account. All of this points to that not all elevation increases, above the ELA, can be assumed to be from snow accumulation. Therefore, we revise the parameterization to take the horizontal ice flow velocity into account. Using all those considerations we define the following density parameterization:

$$\tilde{\rho} = \begin{cases} \rho_i \text{ , if v}_{\text{surf}} \geq 30 \text{ m year}^{-1} \text{ and } \frac{d\tilde{H}}{dt} \leq 0 \\ \rho_i \text{ , if H} \leq \text{ELA} \\ \rho_i \text{ , if dynamical ice build-up is known} \\ \rho_s \text{ , elsewhere.} \end{cases} \quad (5)$$

Below the ELA we assume that the appropriate density is that of ice ($\rho_i$ = 917 kg m$^{-3}$) as it is assumed that all snow melts during the melting season. In areas above the ELA that have a horizontal velocity $v_{\text{surf}}$ greater than 30 m year$^{-1}$, and a negative $\frac{d\tilde{H}}{dt}$, we assume that the negative SEC is caused by ice dynamics, and hence assign $\tilde{\rho} = \rho_i$. This choice of the horizontal velocity threshold is based on the work of Nilsson et al. (2022). In areas with a positive $\frac{d\tilde{H}}{dt}$, that are known to have dynamical ice build-up, $\tilde{\rho} = \rho_i$ is also used. For the rest of the places, we assume snow densities, which are estimated by the firn model. To determine the area of dynamic build-up of ice, we compute the divergence of ice flow from ice velocity. Areas that have a larger inflow than outflow have a negative divergence (build-up). The ice flow velocities from Greenland are from Nagler et al. (2015), and the ice velocity data for Antarctica are from Rignot et al. (2011) and Mouginot et al. (2012). Values of $\tilde{\rho}$ are shown in Fig. 3C and D for Greenland and Antarctica, respectively. To get ice sheet-wide SEC results for AIS (Fig. 4B and D) we used the monthly gravity solutions from GRACE-FO created by Watkins et al. (2015), to fill up the polar gap from ICESat-2.





The gravity solutions were given in liquid water equivalent, which we weighted with the modelled snow densities to get the

SEC estimate in the polar gap.

The uncertainty on $\frac{d\tilde{H}}{dt}$ is obtained as the quadratic sum of the individual uncertainties of the SEC signal, vertical bedrock movement, and the firn compaction. Then the $\frac{d\tilde{H}}{dt}$ uncertainty is summed up for each basin. Finally, assuming that on a large scale the uncertainties are statistically independent, the ice sheet-wide uncertainty is computed by summing all the basin uncertainties quadratically (see Tab. 1).



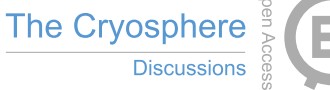

**Figure 3.** Firn compaction rate $\frac{dh_{air}}{dt}$ according to Eq. 2 (A, B), spatial density calculated from Eq. 5 (C, D) and vertical bedrock velocity (E, F) for GrIS and AIS. The white areas in C and D correspond to the ice density, i.e. $\tilde{\rho} = \rho_i = 917 \, \mathrm{kg \, m^{-3}}$. The gray area around Antarctica represents the ice shelves. Values in A, B, E, and F are in units of ice equivalent. The blue area in F are small negative values close to zero.





## 3   Results

For GrIS, we observe a volume change of -279.3±21.0 km$^3$ year$^{-1}$ based on ICESat-2 data, with the largest signals seen around the major outlet glaciers. Over the interior of the GrIS we see a positive SEC in the northeastern part, and a negative signal in the western part, (Fig. 2A and B and Tab. 1). The impact on volume changes from changes in the GrIS firn compaction rate is modelled to be -23.1 ±4.5 km$^3$ year$^{-1}$, with the largest negative signals along the southeastern margin and smaller positive signals in the central east and northwest region (see Fig. 3A and Tab. 1). The modelled vertical bedrock velocities over the GrIS are between 0 and 2 mm year$^{-1}$ in the interior and up to 22 mm year$^{-1}$ near the large outlet glaciers. Summed up over the GrIS the vertical bedrock movement accounts for a volume change of 8.10±0.01 km$^3$ year$^{-1}$ (Fig. 3E and Tab. 1). Figure 4 shows the corrected $\frac{d\tilde{H}}{dt}$ for the GrIS (A), and the associated errors (C). We note the large corrected volume loss in the proximity of the larger outlet glaciers all over Greenland, including Sermeq Kujalleq (Jakobshavn Isbræ, in the central west), Rink, Hayes, Upernavik and Heilprin glaciers (northwest), Petermann glacier (north), Zachariae Isstrøm and Nioghalfjerdsfjorden (79N) glacier (northeast), Kangerlussuaq glacier (central east), and Helheim (southeast). When combined with the appropriate densities (Eq. 5) and evaluated over the entire ice sheet, we estimate that GrIS lost 237.5±14.0 Gt year$^{-1}$ in the period from 2018 to 2021, with the majority of the mass loss occurring at altitudes below 2000 m (see Tab. 1).

For the grounded AIS, we observe a volume change of -42.9±54.0 km$^3$ year$^{-1}$, with the majority of the interior showing minor SECs (±5 cm year$^{-1}$). Larger signals are found at the Kamb Ice Stream (WAIS), with >0.5 m year$^{-1}$ positive elevation change (increased elevation). The largest negative elevation changes (decreased elevation) are detected at the outlet glaciers of the Amundsen Sea sector (Thwaites, Pine Island, and Getz glaciers) and Totten glacier in the EAIS (see Fig. 2B and D, and Tab. 1). The change in firn compaction is within ± 5 cm year$^{-1}$ for the majority of the AIS, and only around the edge of the grounded AIS are larger compaction rates obtained. In particular, over the Antarctic Peninsula, around the Ronne-Filcher ice shelf, and Wilkes Land and George V Land in East Antarctica, we obtain negative compaction rates. Around Princess Elizabeth Land, Amery ice shelf, EAIS, and Marie Byrd Land, WAIS, we obtain larger positive rates. Integrated over the grounded AIS, the firn compaction corresponds to a volume change of -46.7 ±9.4 km$^3$ year$^{-1}$ (see Fig. 3B and Tab. 1). Regarding the vertical bedrock movement, there are a few AIS regions where we expect a downward movement, depicted in blue in Fig. 3F. For the rest of the AIS, the bedrock movements are in the range 0-2 mm year$^{-1}$; however, in WAIS the movement reaches the 25 mm year$^{-1}$ level. Summed up the vertical movement corresponds to a volume change of 21.80±0.03 km$^3$ year$^{-1}$ over the grounded AIS.

Figures 4B and D show the $\frac{d\tilde{H}}{dt}$ for the AIS, and the associated errors. There are large (>0.75 m year$^{-1}$) positive signals at the Kamb ice stream, the windward side of the AP, and smaller regions near the coast of Coats Land and Dronning Maud Land. Whereas, the largest negative signals show a corrected SEC of more than -3.5 m year$^{-1}$ (Thwaites, Pine Island, and Getz glacier). Integrated over the grounded ice, we find that AIS has lost 135.7±27.3 Gt year$^{-1}$, with WAIS being the largest contributor. Along the coast of AIS very local positive SEC signals are observed (see Fig. 4B), which are due to ice-rises and rumples. These are local features that are grounded to the bedrock and are typically located right on the ice sheet/ice shelf





border, and they exist all around the coast (MacAyeal et al., 1987; Matsuoka et al., 2015). They are important for the ice sheet due to their buttressing abilities (Goel et al., 2020).

**Figure 4.** Corrected SEC (A,B) and their associated errors (C, D) for GrIS and AIS. All are expressed in metres of ice equivalent. For AIS, the polar gap has been filled out with GRACE-FO data. Note that the color scale for the SEC is non-symmetric with respect to zero.





**Table 1.** Volume change, SEC corrections due to firn compaction and vertical bedrock velocity, and change in mass balance integrated over the ice defined by Zwally et al. (2012). All results are shown for the total ice sheet, below and above 2000 metre altitude for Greenland and divided into East, West, and AP for Antarctica.

|  | | Greenland | | | Antarctica | | | |
|---|---|---|---|---|---|---|---|---|
|  | Unit | Total | >2000m | <2000m | Total | AP | West | East |
| ICESat-2 | [km³/yr] | -279.3±21.0 | -19.3±7.4 | -260.1±13.6 | -42.9±54.0 | 11.0±12.9 | -103.4±22.9 | 49.5±47.2 |
| Firn corr. | [km³/yr] | -21.5± 4.5 | -14.8±2.4 | -6.7±2.2 | -46.7±9.4 | -17.1±2.2 | -1.5±6.0 | -28.0±6.9 |
| Vertical corr. | [km³/yr] | 8.10±0.01 | 3.29±0.00 | 4.81±0.01 | 21.80±0.03 | 0.80±0.01 | 11.22±0.02 | 9.78±0.02 |
| MB | [Gt/yr] | -237.5±14.0 | -18.7±3.6 | -218.8±10.7 | -135.7±27.3 | 1.5±6.7 | -123.7±13.1 | -13.5±23.0 |

Figure 5 shows the integrated mass balance on basin scale for Greenland and Antarctica. Greenland has eight basins and Antarctica has 27 basins according to the Zwally et al. (2012) definition. Figure 5A shows that all eight basins of the GrIS have a negative basin mass balance, with basins 2 (northeast) and 5 (south) losing the least mass. Basin 8 (northwest) has the largest mass loss of 62.4±6.4 Gt year[-1], which corresponds to 26.3% of the ice sheet-wide mass loss. In Antarctica, 18 out of 27 basins show a positive mass balance, with basin 18 (midwest) having the largest positive mass balance of 22.8±2.4 Gt year[-1]. The

basin with the largest mass loss in Antarctica is basin 21, which has lost 68.6±5.5 Gt year[-1], corresponding to 50.7% of the total mass balance.

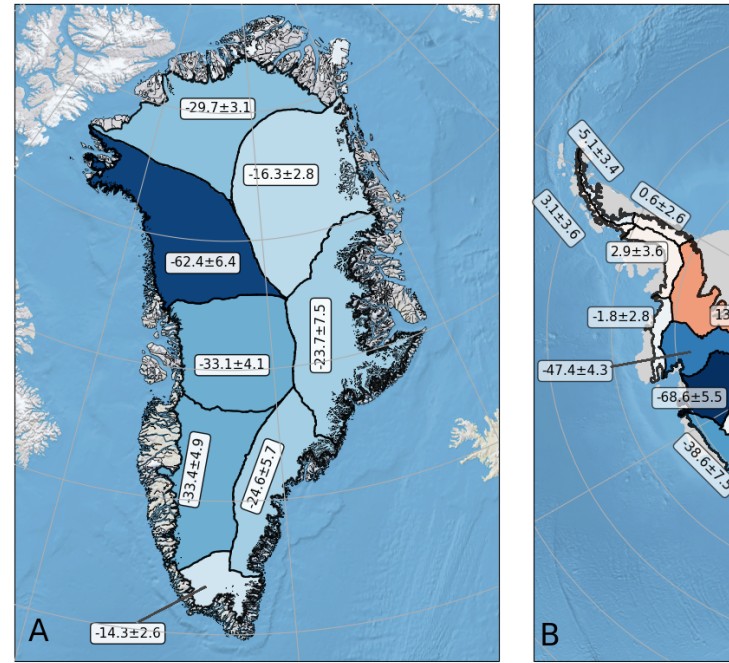

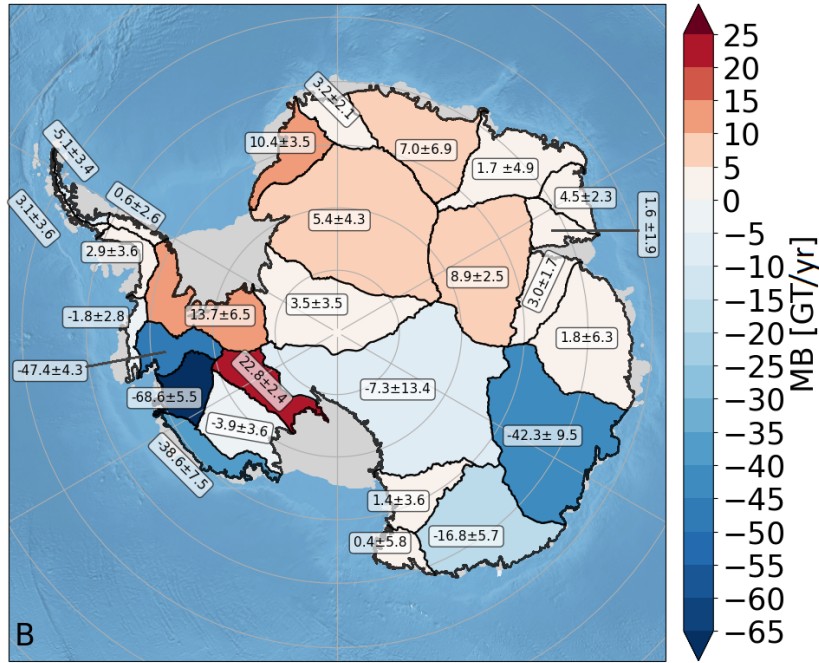

**Figure 5.** Mass balance integrated over the GrIS and AIS basins defined by Zwally et al. (2012). Labels express the mass balance for each basin in units of GT per year. Note that the color scale is non-symmetric with respect to zero.



## 4 Discussion

All regions of both ice sheets, with the possible exception of the Antarctic Peninsula, show a negative mass balance once the firn and vertical corrections are applied (Tab. 1). The combined mass loss from AIS and GrIS is -373.2 $\pm$ 41.3 Gt year$^{-1}$ (or 1.03 $\pm$ 0.11 mm global sea level rise per year, assuming a conversion factor of 362.5 Gt of ice is equivalent of 1 mm sea level rise (Zemp et al., 2019)), using the basins delineated by Zwally et al. (2012). It should be noted that the results only include contributions from the main ice sheet, while peripheral glaciers are excluded. We therefore neglect the mass loss occurring at some larger ice-covered areas, such as Berkner Island, which is surrounded by the Filchner-Ronne Ice Shelf in Antarctica, and Flade Isblink ice cap in Northeast Greenland. There are still significant ambiguities when it comes to separating peripheral glaciers from the ice sheets, which makes it difficult to compare the mass balance if included (Hock et al., 2023) and we therefore focus only on the grounded areas of the two ice sheets. If the periphery glaciers were included, we would most likely obtain a higher mass loss for both Antarctica and Greenland.

To compare our results with other studies, Mankoff et al. (2021) estimate the GrIS mass balance to be -282.0 $\pm$ 92.0 Gt year$^{-1}$ for the same period, and the ESA climate change initiative (CCI) gravimetric mass balance estimate from GRACE-FO is -296.7 $\pm$ 54.24 Gt year$^{-1}$ using methods described in Barletta et al. (2013, 2021). For the grounded AIS the CCI gravimetric mass balance from GRACE-FO is -112.6 $\pm$ 84.3 Gt year$^{-1}$ (Groh et al., 2019; Groh and Horwath, 2021), both the mass balance estimates for GRACE-FO cover the period from October 2018 to August 2021. Our estimates are therefore less negative but within the uncertainties for GrIS. In general, estimates using the input-output method (e.g. Mankoff et al. (2021)) have proven to give higher mass loss estimates than the altimetry method (Otosaka et al., 2023). As GRACE-FO has a coarse resolution, the gravimetric mass balance estimate will include a leakage error in the mass signal, caused by nearby surrounding peripheral glaciers and ice caps and also ice-covered areas e.g. northeast Canada, which are not a part of the ice sheet. This is especially a problem in Northern Greenland (Baur et al., 2009; Barletta et al., 2013). Similarly, in Antarctica, the surrounding ocean creates a leakage error (Horwath and Dietrich, 2009; Chen et al., 2015).

Figure 4C and D, show that the errors on our estimate of the SEC signal are low over the interior regions for both GrIS and AIS due to the relatively flat and homogeneous surface inland. The largest absolute uncertainties are found in areas with steep and complex terrain. For Greenland, the largest uncertainties (>0.25 m year$^{-1}$) are found in the southeast region, while for Antarctica the largest uncertainties (>0.25 m year$^{-1}$) are located at the Antarctic Peninsula, the Trans-Antarctica Mountains, and Oates Land.

Our results show that correcting for the appropriate surface density, $\tilde{\rho}$, (Eq. 5), results in a non-linearity in the volume-to-mass conversion, as evident in Tab. 1. Other studies have used a constant density over the entire ice sheet. For example, Smith et al. (2020) used ice densities for all zones of both AIS and GrIS. This may overestimate the mass balance in areas of snow-induced SEC, most evident in the EAIS. For Greenland, other studies assumed a density parameterization based on the location, where the ice sheet has been divided into accumulation and ablation zones using the density of snow, $\rho_s$, and ice, $\rho_i$, respectively (Thomas et al., 2006; Sørensen et al., 2011; Zwally et al., 2021). This is not appropriate in Antarctica as it neglects





the ice dynamic signal (positive from ice build-up) in the accumulation area, across almost the entire grounded AIS. With the
updated parameterization of $\tilde{\rho}$ (see Eq. 5) we can provide a valid solution applicable to both ice sheets.

The SEC signal includes a component attributable to basal mass balance (see Eq. 2). Basal mass balance includes basal
melt generated by three heat components: geothermal heat, frictional heat, and heat from surface meltwater penetrating to the
bed (Karlsson et al., 2021). The basal melt created by geothermal heat and frictional heating leads to surface lowering and it

is therefore included in the observed SEC. However, the generation of basal melt through surface meltwater energy usually
occurs in localized conduits (Karlsson et al., 2021). It is therefore likely that basal melt originating from surface meltwater will
remain undetectable in the SEC signal. For the GrIS, Karlsson et al. (2021) estimates that basal melt production is around $21\pm4$
Gt year$^{-1}$, where geothermal heating and the frictional heat are responsible for around $16\pm5.5$ Gt year$^{-1}$, and surface meltwater
is responsible for the rest. This means that the majority of the basal melt will be captured in the SEC signal for GrIS. Observed

basal melt rates are sparse over the grounded part of Antarctica but on the basis of geothermal models, Jordan et al. (2018)
found basal melt rates of up to 6 mm year$^{-1}$ near the South Pole and Fisher et al. (2015) found melt rates of up to 18 mm year$^{-1}$
in WAIS. More recently, Artemieva (2022) created a new geothermal model and concluded that previous geothermal models
have significantly underestimated the basal melt rates. Furthermore, Colgan et al. (2021) found that the geothermal heat flux for
both ice sheets must be considered at sub-kilometre horizontal scales to account for the full effect of the subglacial topography.

Consequently, although some of the identified SEC likely includes basal mass balance, it is still challenging to quantify this in
Antarctica.

The high spatial and temporal resolution of the SEC data allows us to infer drivers of the mass balance in light of surface
mass balance (SMB) and ice dynamic processes. For instance, in Antarctica SMB in some basins varies strongly according to
the SAM. The SAM index describes the relative position and intensity of westerly winds around the continent and is related

to other southern hemisphere climate patterns such as the Amundsen Sea Low and the El Nino Southern Oscillation (ENSO),
as well as ozone depletion (Fogt and Marshall, 2020). The SAM has a positive and a negative phase with opposing impacts on
SMB in different basins around Antarctica (e.g. Hansen et al. (2021)). The trend of the SAM index has been generally positive
since the 1980s (Fogt and Marshall, 2020), including in the period of this study (0.37) (Marshall, 2023). Hansen et al. (2021)
showed that a positive SAM phase gives a positive SMB anomaly in three of the four Peninsula basins (basins 24,25,27 in

Hansen et al. (2021)) and a negative SMB anomaly in the last Peninsula basin (26). This fits with our results over the Peninsula
and suggests that recent total MB trends over the Peninsula have mainly been influenced by the positive SMB. Likewise,
Hansen et al. (2021) also found that a positive SAM phase gives a positive SMB anomaly in Zwally et al. (2012) basins 1-5
and 7, (the Weddell Sea and Dronning Maud Land sector), which again correlates with the mass gain in our results. As with the
importance of the SAM in Antarctica, precipitation in southeast Greenland is strongly related to the position of the Iceland low

(Berdahl et al., 2018) showing that both ice sheets are influenced strongly by the regional circulation patterns. Understanding
these remains is crucial for understanding the present-day mass change and future evolution.

Widespread summer melt over western and northern GrIS, particularly as observed in July 2019 (Sasgen et al., 2020), is
likely the cause of the negative SEC in these regions of the GrIS. We note also the prominent SEC signal detected around the



large outlet glaciers at Sermeq Kujalleq, Helheim, Upernavik and Kangerdlussuaq, suggests in these regions ice dynamics are
the dominant driver of SEC (Fig. 4A).

## 4.1 Decadal mass changes in Greenland

Ice sheet mass balance responds slowly to climate change and the study of Sørensen et al. (2011) presents the opportunity to
investigate how the mass balance has changed spatially over the GrIS between the ICESat (2003-2008) and ICESat-2 (2018-
2021) periods. This data set is only available over GrIS and both data sets are given in mass balance derived with slight
differences in the applied appropriate density ($\tilde{\rho}$) (Sørensen et al. (2011) used for ICESat $\rho_s$ for positive SEC signals and $\rho_i$ for
negative signals, and for ICESat-2 we have used Eq. 5). Figure 6 shows the difference between the ICESat-2 and ICESat mass
balances, defined as $\Delta MB = \frac{dM}{dt_{IS}} - \frac{dM}{dt_{IS2}}$.

In Fig. 6, positive values of $\Delta MB$ are resulting from an increasing mass loss from the 2000s to the 2020s while negative
values of $\Delta MB$ mean that there has been a decrease in mass loss from the ICESat period to the ICESat-2 period. This
comparison allows us to identify the importance of regional circulation and extreme events in both atmosphere and ocean
to changes in ice sheet mass changes. Hence, Fig. 6 reveals that the thinning of Jakobshavn glacier has been slowing down
between the two periods, which is also found in other studies e.g. (Khazendar et al., 2019; Joughin et al., 2020). This decrease in
mass loss is likely due to a decrease in the ocean temperature in Disco Bay leading to reduced ocean forcing (Khazendar et al.,
2019). There are also glaciers along the southeast coast that show a decrease in mass loss between the two periods. Indeed, as
pointed out by Murray et al. (2010) and Seale et al. (2011), in the early 2000s there was an increase in ocean temperature along
the coastal currents in the southeast, which sped up the dynamic mass loss in that region; this has been followed by a cooling
in ocean temperatures in the late 2000s which slowed down the thinning of the southeast glaciers.

In the northeast we see that the mass loss has increased in the entire sector, with the largest differences from the Zachariae
Isstrøm and Nioghalvfjerds glacier; in the northeast sector, the increase in mass loss is primarily due to an increase in discharge
(Mouginot et al., 2015, 2019). Whereas, the increase in mass loss in the northwest emerges from both increase in runoff and
discharge (Mouginot et al., 2019; Black and Joughin, 2022). Other studies have found that the northernmost part of the GrIS
has experienced a major increase of the ablation zone area, due to the cloud cover increasing in the spring/early summer and
enhanced atmospheric warming through a decrease in longwave heat loss (Noël et al., 2019) and thus increases the runoff.





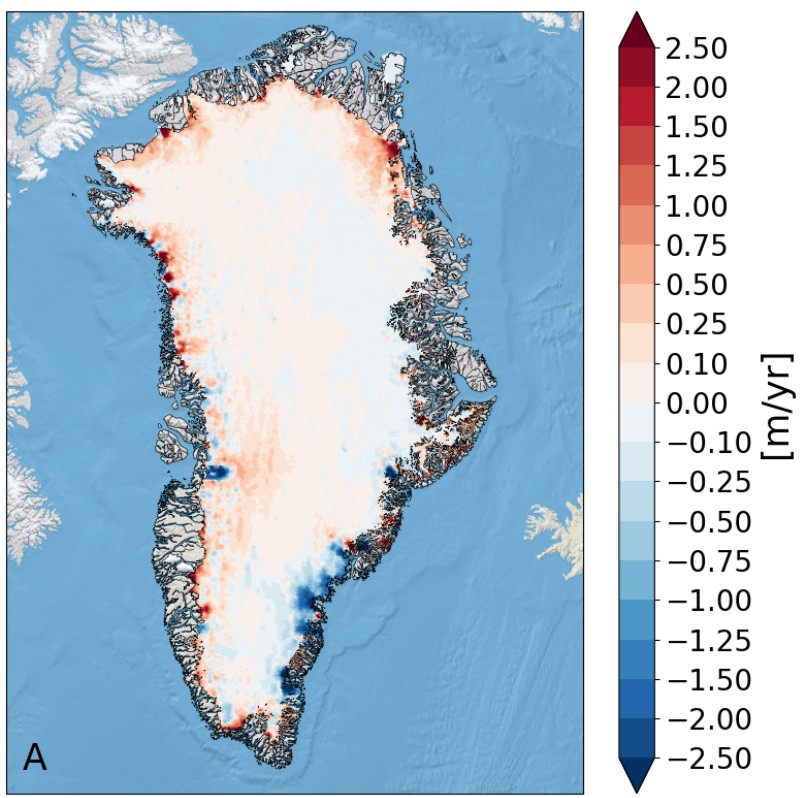

**Figure 6.** Difference between the mass change from ICESat-2 (2018-2021), this study, and results for the ICESat (2003-2008) obtained by Sørensen et al. (2011), units are in metres of ice equivalent.

Overall, Fig. 6 shows that the mass loss has moved further north. Mattingly et al. (2023) attribute some of this enhanced mass
loss in northern Greenland to atmospheric rivers that drive foehn winds, particularly in the north and northeast. Atmospheric rivers are long streams of warm moist air transported from the tropics to higher latitudes by large-scale atmospheric circulation patterns and are associated with intense precipitation over both ice sheets. As well as heavy rain and/or snowfall leading to large mass accumulation, the transport of warm moist air is also associated with melt events over lower elevation regions driven both by foehn winds and high sensible heat fluxes. Their impact on the mass balance of ice sheets is therefore on a local to regional
scale and can be complex including both mass gain at higher elevations and mass loss at lower elevations. With a warming climate, the rise in atmospheric moisture is anticipated to amplify the influence of atmospheric rivers on the ice sheet's mass balance. Our analysis indicates that this impact might already be noticeable in SEC data. Davison et al. (2023) have found that short-term atmospheric rivers have had an impact on the yearly mass balance in West Antarctica. Our analysis shows that the pattern of ice loss over both polar ice sheets is consistent with climate change and the basic processes that control ice sheet
surface mass balance and dynamics.



# 5 Conclusions

We employed ICESat-2 data to investigate the SEC over the GrIS and grounded AIS, in the period from October 2018 to September 2021. From the observed SEC we obtained mass changes by correcting for firn compaction and vertical bedrock movements. We have updated the description of the appropriate density needed for the volume-to-mass change conversion,

obtaining a parameterization that is valid for both ice sheets. In particular, we take into account the sign of the observed SEC and pinpoint areas of dynamical ice build-up on the basis of ice flow velocities. We find that the grounded AIS has lost $135.7\pm27.3$ Gt year$^{-1}$ and the GrIS has lost $237.5\pm14.0$ Gt year$^{-1}$. The mass loss in Antarctica is driven mainly by West Antarctica and a few glaciers in the East, whereas the parts of the Antarctic Peninsula and East Antarctica are experiencing a small mass gain, likely related to the phase in the SAM (Hansen et al., 2021). Conversely, the majority of the Greenland ice sheet

is experiencing mass loss. Our mass balance estimates are in good agreement with estimates from other studies, especially for Antarctica, but we emphasize the importance of accurately representing the density of snow and firn. By comparing our results with the Greenland mass balance obtained by Sørensen et al. (2011) from ICESat observations for 2003-2008, it is evident that the trend of mass loss over GrIS has been maintained overall but the pattern of ice loss has changed and migrated northwards and further inland in recent years. The mass loss in Antarctica is also partly masked by interannual and multiannual climate

variability but our results suggest the onset of similar patterns of ice sheet change to those already observed in Greenland. Suggesting that similar Earth system processes are driving change over both ice sheets.

*Data availability.* The mass balance estimates are both available at 10.11583/DTU.19500677 data.dtu.dk

*Author contributions.* NH, SBS and LSS conceived the study and processed the ICESat-2 data. DM and GS provided the vertical bedrock correction data and wrote the section on it. NH and RM provided the firn data. NH and SBS wrote the majority of the manuscript with input

from all co-authors.

*Competing interests.* At least one of the (co-)authors is a member of the editorial board of The Cryosphere.

*Acknowledgements.* The present work is a contribution to the EU Horizon 2020 PROTECT project, contribution number XX. GS is funded by a DIFA RFO grant. DM is funded by a INGV 2020-2023 "Ricerca libera" research grant and by the MACMAP INGV Departmental Project. RM also acknowledges the support of the Danish Government via the National Centre for Climate Research. This study was partially

funded through the CCI+ Phase 1 Greenland_Ice_Sheet_cci project (ESA contract No. 4000126523/19/I-NB). Finally, we want to thank Bert Wouters for his suggestions on how to improve the manuscript.



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
