# Peer review of "Revisiting ice sheet mass balance: insights into changing dynamics in Greenland and Antarctica from ICESat-2"

_The Cryosphere, 2023_

## Referee Comment (RC1)

**1   Overview**

Hansen et al. [2023] use data from NASA's ICESat-2 mission to derive volume and mass change estimates for Greenland and Antarctica. They build upon the methods of Sørensen et al. [2011] to derive 5km along-track segments of elevation change, which are interpolated to regions of grounded ice. While the work presented by the authors falls within the scope of *The Cryosphere*, I am skeptical of the results and the methods as presented in the current version of the manuscript. There are several issues that should be resolved before I would recommend the publication of this work.

**2   Major comments**

- The ICESat-2 time series is short and the interpretation of changes can be heavily dependent on the period of study. Interpreting a 3-year period (2018–2021) in the context of a 6-year ICESat-2 study (2003–2009) can lead to spurious conclusions. The Antarctic time series is particularly sensitive to variability in snowfall. I believe that the trends presented here are too short for putting in the long-term context discussed in the manuscript.

- Fitting along-track ICESat-2 data with a 5km plane will likely lead to poor results. I would recommend comparing your (uncorrected) volume change estimates with the ATL15 gridded elevation change product.

- The conversion of mass to density is non-trivial. The zone in Greenland assumed to be pure ice change is particularly large and spans into the interior. Something seems off in this conversion as presented here. I think following methods outlined in e.g. Smith et al. [2020] to remove the variability in firn air content (FAC) may lead to better results for the residual mass change. There have been some improvements in firn and surface process modeling since the publication of Sørensen et al. [2011].

**3   Minor comments**

- Some awkward grammar or phrasing throughout the manuscript

- Some uses of "Furthermore", "Moreover", "Therefore" seem superfluous

**4   Line-by-line comments**

**Page 1, Line 1:** The NASA GEDI mission is slightly newer (launched December 2018). I would change this from "latest satellite laser altimetry" to "satellite laser altimeter"

**Page 1, Line 2:** "mature enough"

**Page 1, Lines 12–13** : These sentences use very similar language as in the title of Smith et al. [2020] "Pervasive ice sheet mass loss reflects competing ocean and atmosphere processes". Might want to include a citation.

**Page 1, Lines 16–17** : Why is the result of mass loss from land ice "in spite of their contrasting geographic contexts"?

**Page 1, Line 19** : Similar as above, might want to cite Smith et al. [2020].

**Page 2, Line 42** : This sentence is particularly long. Could split it before "other outlet glaciers".

**Page 2, Lines 43–47** : This sentence is also particularly long.

**Page 3, Lines 56–58** : Except in a possible melt context, I am not sure if comparing Greenland and a future Antarctica in this way is right. The Peninsula may evolve similar to Greenland, but West Antarctica will likely evolve in a MISI context. East Antarctica is still a question (and may be different between drainages).

**Page 3, Line 58** : Remove "therefore"

**Page 3, Line 84** : I would change this to be similar to "measures the two-way return time of the laser beam with photon-level precision". The use of "times" could be interpreted as "multiplies".

**Page 3, Line 85** : Latest estimates of the laser footprint is closer to 11m [Magruder et al., 2020].

**Page 3, Line 86** : Because the segments overlap, I believe that the posting should be 20m with an along-track segment length of 40m.

**Page 4, Function 1** : Concerned about the impact of non-linear features at the 5km segment length. The profile of the ice sheet towards the periphery can not be well approximated by a plane at this segment length.

**Page 4, Lines 103–104** : Doesn't function 2 describe that the elevations are corrected for non-ice mass related processes?

**Page 6, Figure 2** : Please don't use *jet* as a colormap

**Page 9, Line 169** : what about snow melt?

**Page 9, Line 176** : van den Broeke

**Page 9, Line 189** : how was your ELA estimated?

**Page 9, Line 193** : was the snow density estimated or parametrized in the firn model?

**Page 9, Line 198** : how different is the gravity solution to the altimetry solution for change? How were errors considered here?

**Page 12, Line 230** : should split this into the elastic and GIA related components.

**Page 14, Figure 5** : except for visualizing the basin delineations, this would work better as a table.

**Page 15, Line 248** : I am confused by this statement as Figure 5 shows regions of mass gain in Antarctica.

**Page 15, Lines 258–267** : With such a short time series, these comparisons are especially sensitive to the period of interest (see figure on next page)

[Figure]

Figure 1: Changes in Antarctic ice mass from GRACE-FO for the period of interest in the study (2018–2021), and extended into 2023

**Page 15, Line 268** : Comparisons with GRACE-FO for this short of a time series are also sensitive to uncertainty in the accelerometer transplant over 161-day cycles.

**Page 15, Line 276** : The description of the methods of Smith et al. [2020] is incorrect. They used time-variable firn air content (an estimate of porosity variability) from **?** to convert from elevation change to ice-equivalent elevation change.

**Page 17, Line 317** : Do ice sheets actually respond slowly to climate change?

**Page 17, Lines 223–224** : I don't believe that the shortness of this trend allows this mass change estimate to be placed in context of climate change. It is highly sensitive to short-term variability.

**References**

N. Hansen, L. S. Sørensen, G. Spada, D. Melini, R. Forsberg, R. Mottram, and S. B. Simonsen. Revisiting ice sheet mass balance: insights into changing dynamics in Greenland and Antarctica from ICESat-2. *The Cryosphere Discussions*, 2023:1–28, 2023. doi: `10.5194/tc-2023-104`.

L. A. Magruder, K. M. Brunt, and M. Alonzo. Early ICESat-2 on-orbit Geolocation Validation Using Ground-Based Corner Cube Retro-Reflectors. *Remote Sensing*, 12(3653), 2020. ISSN 2072-4292. doi: `10.3390/rs12213653`.

B. Smith, H. A. Fricker, A. S. Gardner, B. Medley, J. Nilsson, F. S. Paolo, N. Holschuh, S. Adusumilli, K. Brunt, B. Csatho, K. Harbeck, T. Markus, T. Neumann, M. R. Siegfried, and H. Zwally. Pervasive ice sheet mass loss reflects competing ocean and atmosphere processes. *Science*, 2020. ISSN 0036-8075. doi: `10.1126/science.aaz5845`.

L. S. Sørensen, S. B. Simonsen, K. Nielsen, P. Lucas-Picher, G. Spada, G. Aðalgeirsdottir, R. Forsberg, and C. S. Hvidberg. Mass balance of the Greenland ice sheet (2003–2008) from ICESat data – the impact of interpolation, sampling and firn density. *The Cryosphere*, 5(1):173–186, 2011. doi: `10.5194/tc-5-173-2011`.